# Two-Dimensional Numerical Analysis for TBM Tunneling-Induced Structure Settlement: A Proposed Modeling Method and Parametric Study

**Rashad Alsirawan [1],\*, Ashraf Sheble [2] and Ammar Alnmr [1]**

1   Department of Structural and Geotechnical Engineering, Széchenyi István University, 9026 Győr, Hungary; alnmr.ammar@hallgato.sze.hu
2   Department of Geotechnical Engineering, Tishreen University, Latakia P.O. Box 2230, Syria; ashraf.sheble@tishreen.edu.sy
\*   Correspondence: alsirawan.rashad@sze.hu

**Abstract:** The construction of tunnels in densely populated urban areas poses a significant challenge in terms of anticipating the settlement that may result from tunnel excavation. This paper presents a new and more realistic modeling method for tunnel excavation using a Tunnel Boring Machine (TBM). This method is compared with other reference modeling methods using a validated model of a subsurface tunnel excavated by a TBM with a slurry shield. A parametric study is conducted to investigate the impact of key parameters, including structure width, foundation depth, eccentricity, load on the structure, overburden depth, and tunnel diameter, on tunnel–soil–structure interaction and the resulting structure settlements. The results reveal that the tunnel diameter, eccentricity, and overburden depth have a significant impact on structure settlements, with values of 22.5%, 17%, and 7.1%, respectively. Finally, the paper proposes an equation for predicting the maximum settlement of a structure, considering the critical parameters. The validity of the equation is evaluated by comparing its results with the outputs from various case studies, including a newly validated model, two real-life case studies, and centrifuge tests. The results indicate a high level of consistency between the calculated and measured settlements.

**Keywords:** settlement trough; realistic modeling method; TBM modeling; TBM with slurry shield; parametric study; soil–structure interaction; equation of settlement estimation

## 1. Introduction

The rapid growth of cities, followed by congestion and major traffic challenges, necessitates the extension of existing highways as well as the construction of new roads and bridges. One of the solutions is to exploit the underground space through the construction of tunnel networks. Tunnel excavation is associated with the deformations of the soil surrounding the tunnel called 'volume loss'. These deformations reach the surface, leading to a settlement trough, which causes damage to the surface structures [1]. These soil deformations in the case of Tunnel Boring Machine (TBM) tunneling are created by soil movement towards the excavation face, soil movement in the radial direction resulting from the over-excavating of the soil around the tunnel, and the gap left by the tail of the shield, in addition to the subsequent deformation of the tunnel lining and the consolidation that occurs in the case of clay soils [2].

By adopting a 2D analysis, several methods were presented to model the volume loss around the tunnel, which constitutes the main factor resulting in the surface subsidence caused by tunneling. Vermeer and Brinkgreve [3] hypothesized a two-stage method where the soil within the tunnel is deactivated and the tunnel lining is activated in the first stage. In the second stage, a uniform contraction of the tunnel lining is applied that results in deformation of the surrounding soil towards the inside of the tunnel. Moller [4] also

suggested a two-stage method. In the first stage, the soil within the tunnel is deactivated and a radial pressure is applied on the excavation surface equal to the grout pressure, and with depth it increases linearly by the weight unit of the grout. In the second stage, the tunnel lining is activated and the grout pressure is deactivated. Suwansawat et al. [5] recommended a modification to the grout pressure method for Moller [4], where the volume loss can be modeled in three stages. In the first stage, the soil within the shield is deactivated and the face support pressure is activated. In the second stage, the tunnel lining is activated and the gap between the lining and the shield is filled with soft grout material. Then, the face support pressure within the tunnel is deactivated and replaced by the grout pressure in the gap between the lining and the shield. In the final stage, the grout pressure is deactivated, and the soft grout is replaced with a hardened one.

Numerous parametric studies were conducted on the issue of the interaction between soil, tunneling, and surface structures. Maleki [6] studied the effects of stiffness, weight and geometry of a structure, as well as the eccentricity between the axis of the tunnel and the axis of the structure on the ground surface settlements. Katebi [7] investigated the influence of tunnel depth, structure width and weight, and the eccentricity on the internal forces and deformations of the tunnel lining. Giardina [8] conducted a numerical study on the consequences of the stiffness, the weight of the structure, and volume loss, based on the results of the Farrell [9] centrifuge test. Son [10,11] independently completed two numerical studies of a brick structure that was located over sandy and clayey soils; the researcher studied the effect of changing the volume loss, the depth, and the diameter of the tunnel on the settlement under the brick structure. Boldini [12] conducted an investigation to demonstrate the impact of the number of stories on the settlement trough. Specifically, the study examined how the incremental stiffness of the structure with more stories affects the settlement trough and also investigated the effect of the incremental structure's weight on the settlement trough.

Peck [13] presented an empirical formula to calculate the surface settlement caused by tunneling by adopting the greenfield condition through the normal distribution curve (Gauss). O'Reilly and New [14] proposed an equation to calculate the trough width parameter, which signals the distance between the axis of the tunnel and the point of the maximum slope of the surface settlement trough. Mair et al. [2] assumed that the volume of the settlement trough at the surface is equal to the volume loss around the tunnel in the case of clay soil, and they showed a connection between them based on their assumption. Herzog [15] presented an equation to calculate the maximum settlement above the axis of the tunnel based on experimental results, while Chakeri and Unver [16] developed the work of Herzog [15] and considered the effect of soil cohesion, internal friction angle, and the required supporting pressure on the drilling face of the Earth Pressure Balance shields (EPB). Wang [17] suggested an empirical equation to calculate the subsidence in the case of sandy soil, taking into account the change in the volume of the settlement trough with depth.

The majority of previous studies have primarily focused on the analysis of settlement through greenfield conditions. Furthermore, the parametric study has been limited in its ability to fully comprehend the behavior of settlement under structures, and the proposed equations used to calculate settlements have been primarily based on greenfield conditions. This highlights the need for further research to better understand the complexities of settlement behavior under various conditions, including those that arise when structures are present.

In this paper, a validated model of the subsurface Second Heinenoord tunnel [18] is utilized to calibrate a new and improved modeling method of shield tunneling using the Finite Element Method (FEM). The proposed method is aimed at achieving a more precise and realistic simulation of the tunneling process by accurately representing each of its individual stages. Additionally, an extensive parametric study is conducted to explore the impact of adjacent structure existence on the deformation caused by tunneling in comparison with greenfield conditions and to examine the effects of various critical

parameters on the settlement under adjacent structures. Finally, a new and tailored equation is proposed that can be applied in the elementary analysis to calculate the maximum settlements of adjacent structures.

## 2. Proposed Method: Grout Hardening Method

The proposed 2D modeling method is designed to provide a comprehensive and interconnected understanding of the stages involved in tunnel excavation using TBM with shields. By representing the work progress within the tail of the shield in a manner that closely approximates reality and accounts for the most crucial factors causing volume loss around the tunnel mentioned in Section 1, this method leads to more accurate development of stresses and deformations in the tunnel lining. Additionally, the method enables a more precise modeling of the deformation of the soil around the tunnel and the propagation of this deformation towards the surface, resulting in the creation of a subsidence trough at the surface that aligns closely with the reference field measurements. The tunnel excavation process is systematically divided into four stages as illustrated in Figure 1, allowing for a step-by-step analysis of the changes occurring during the construction process:

1. In the first stage, the soil within the shield is deactivated, and the tail of the shield is activated.

2. In the second stage, the tail of the shield is deactivated, and the grout pressure is activated. This enables us to take into account the radial deformations of the surrounding soil towards the tunnel interior, as a result of the gap produced by the over-excavation of the cutting wheel, the void left by the tail of the shield, and the insufficient grout pressure to resist the soil pressure from above, which is close to the geostatic stresses before executing the tunnel.

3. In the third stage, the grout pressure is deactivated, and the tunnel lining is activated while filling the gap around the tunnel with grout material of an initial stiffness. The stiffness of the grout material is then adjusted until the required volume loss is achieved, resulting in a subsidence trough on the surface that is closest to the reference field measurements.

4. In the last stage, the grout material with initial stiffness is replaced by the grout material with final stiffness. As a result, the radial deformations of the soil towards the interior of the tunnel come to a halt, thereby causing the subsidence trough to cease developing.

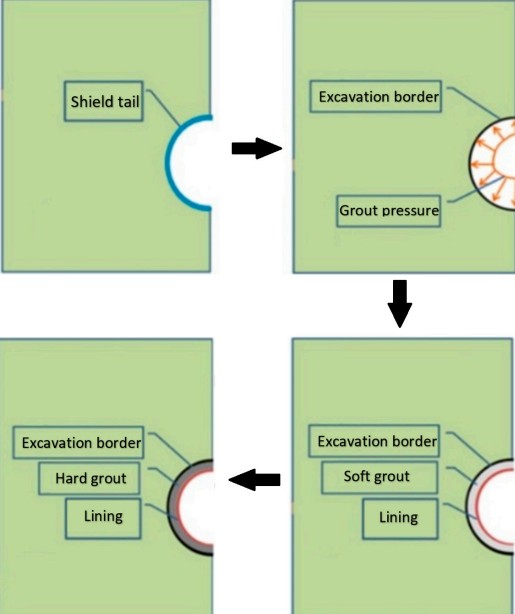

**Figure 1.** Stages of the proposed method (grout hardening method).

## 3. Finite Element Modeling

In order to validate the proposed modeling method, a case study of the subsurface Second Heinenoord tunnel [18] was utilized. This section of the study involves two key aspects: (i) conducting a 2D numerical modeling of the tunnel using the conventional contraction method to determine the appropriate constitutive model, by comparing the predictions of numerical analysis using the Hardening Soil with Small-strain Stiffness (*HSS*) and the Hardening Soil (*HS*), and (ii) comparing the results obtained from three distinct modeling methods (grout pressure method, grout hardening method, and contraction method) with field measurements, with the aim of validating the proposed method.

### 3.1. Description of the Second Heinenoord Tunnel

The Second Heinenoord tunnel was built in the Netherlands in 1996. The outer diameter of the tunnel is $D$a = 8.3 m. A 7-m-long slurry shield with a permanent frontal reinforcement was used to construct this tunnel. The slurry pressure at the top of the shield was 230 kPa that increased linearly with depth by the weight unit of bentonite, which was estimated to be approximately 15 kN/m$^3$. The grout pressure was 125 kPa that increased linearly with depth by the same magnitude. The tunnel lining was made of prefabricated reinforced concrete rings that were 0.35 m thick and 1.5 m long [18].

During the tunneling, several field measurement programs were carried out, including the measuring of the settlements of the ground surface ($S_v$) above the tunnel in a transverse direction perpendicular to the tunnel axis ($X$), as shown in Table 1. The overburden depth ($H$) = 12.5 m from the ground surface to the tunnel upper point, and the groundwater level was 1.5 m. This tunnel is classified as a subsurface tunnel ($H/D < 2$). The ground surface is horizontal. The properties and levels of the soil layers [18] are shown in Table 2.

**Table 1.** Measured transverse settlements [4].

| X (m) | 0 | 6.2 | 8.5 | 10.4 | 16.7 | 20 |
|---|---|---|---|---|---|---|
| $S_v$ **(mm)** | 26.2 | 16.86 | 10.44 | 6.96 | 1.34 | 0.34 |

**Table 2.** Properties of soil layers surrounding the tunnel.

| Layer | Depth (m) | $\gamma_{sat}$ (kN/m$^3$) | ∅ (º) | $c'$ (kPa) | ν |
|---|---|---|---|---|---|
| Fill | 0–4 | 17.2 | 27 | 3 | 0.34 |
| Sand | 4–19.75 | 20 | 35 | 1 | 0.3 |
| Sand | 19.75–23.25 | 20 | 35 | 1 | 0.3 |
| Sand-Clay | 23.25–27.5 | 20 | 31 | 7 | 0.32 |

$\gamma_{sat}$: saturated unit weight, ∅ : internal friction angle, $c'$ : cohesion, ν: Poisson's ratio.

For the purpose of validation, the field-measured settlements of the ground surface during the construction of the Second Heinenoord tunnel had been compared to the calculated settlements using the 2D numerical model of the tunnel. During this process, two constitutive models, the Hardening Soil with Small-strain Stiffness (*HSS*) and the Hardening Soil (*HS*), and three modeling methods had been applied, which are the grout pressure method [19], the contraction method [20], and the proposed method (grout hardening method).

### 3.2. Two-Dimensional Geometric Model

The continuous approach was implemented to model the tunnel and the surrounding soil layers. Due to symmetry, only half of the model was simulated, with the geometric dimensions of the 2D numerical model appropriately chosen to meet the German requirements [21]. Figure 2 shows the geometric dimensions of the model as well as the boundary conditions, i.e., horizontal transitions are prohibited in the side boundaries, while vertical and horizontal transitions are prohibited in the lower boundaries. Rotating and horizontal transitions are also prohibited at tunnel lining nodes located on the symmetry axis. To

determine the appropriate mesh size, a sensitivity analysis was conducted on the mesh. The simulation utilized a coarse mesh distribution with local refinement at the surface line and tunnel line since the settlement behavior demonstrated little difference between coarse and medium distributions. A mesh of the model had 317 trigonometric elements, each of which consisted of 15 nodes.

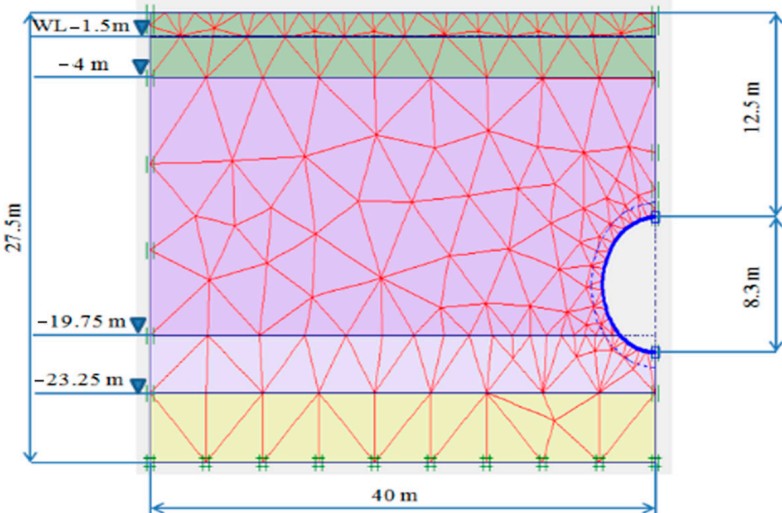

**Figure 2.** Finite element mesh of the 2D model.

In order to account for the impact of the joints between the prefabricated concrete parts, the tunnel lining was modeled using plate elements with linear elastic behavior, while the bend stiffness was reduced by dividing it by a reduction coefficient equal to four [22]. Table 3 presents the characteristics of the lining.

**Table 3.** Lining properties.

| Constitutive Model | $\gamma$ (kN/m$^3$) | $EA$ (GN/m) | $EI$ (MNm$^2$) | $\nu$ | $t$ (cm) |
|---|---|---|---|---|---|
| Linear elastic | 24 | 10.5 | 26.78 | 0.15 | 35 |

$\gamma$: unit weight, $EA$: axial stiffness, $EI$: flexural stiffness, $t$: thickness of lining.

3.2.1. Appropriate Constitutive Model

The proper validation of the problem is one of the most important issues to be taken into consideration. In order to validate a model, the outputs of a numerical method are assessed through a comparison with the field measurements or the results of software packages that employ independent solutions [23]. The first crucial step in geotechnical numerical modeling is selecting and calibrating a robust and accurate soil material model (i.e., constitutive model). A less precise model may yield poor results and make estimating the proper behavior in the field more complex [24].

The reference studies suggested the *HS* and the *HSS* models. Therefore, the authors carried out a comparative study to simulate the behavior of the soil utilizing the contraction method. Table 4 illustrates the parameters of both *HS* and *HSS* models applied in our study. Compared with the field measurements, the *HSS* model yields the closest shape to the transverse settlement trough as shown in Figure 3, which is consistent with the results of Law [25]. Figure 4 demonstrates the vertical displacements in soil that are caused by the contraction method and the *HSS* model. After having determined the optimal constitutive model (*HSS*) that simulates the soil behavior more accurately, a comparison of the predictions of the three modeling methods described above was conducted.

**Table 4.** Parameters of *HS* and *HSS* models.

| Layer | $v_{ur}$ | $E_{oed}^{ref}$ (MPa) | $E_{50}^{ref}$ (MPa) | $E_{ur}^{ref}$ (MPa) | $G_0^{ref}$ (MPa) | $\gamma_{0.7}$ (%) | $m$ (-) | OCR | $R_{inter}$ |
|---|---|---|---|---|---|---|---|---|---|
| 1 | 0.2 | 14 | 14 | 42 | 52 | 0.0005 | 0.5 | 1 | 1 |
| 2 | 0.2 | 35 | 35 | 105 | 175 | 0.0005 | 0.5 | 1 | 0.67 |
| 3 | 0.2 | 35 | 35 | 105 | 175 | 0.0005 | 0.5 | 1 | 0.67 |
| 4 | 0.2 | 7 | 12 | 35 | 88 | 0.0005 | 0.9 | 1 | 1 |

$v_{ur}$ : unloading/reloading Poisson's ratio, $E_{oed}^{ref}$: reference tangent stiffness, $E_{50}^{ref}$ : reference secant stiffness, $E_{ur}^{ref}$ : reference unloading/reloading stiffness, $G_0^{ref}$ : reference shear modulus at small strain, $\gamma_{0.7}$ : reference strain threshold, $m$: exponential power, *OCR*: over consolidation ratio, $R_{int}$ : strength reduction factor.

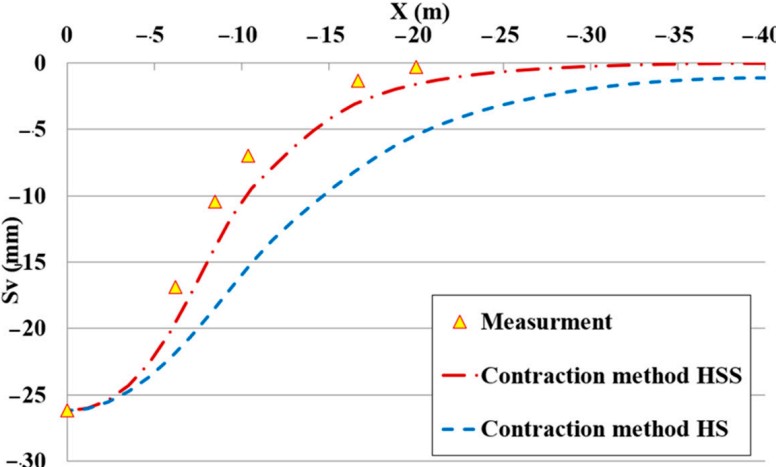

**Figure 3.** Effect of *HS* and *HSS* on the settlement trough.

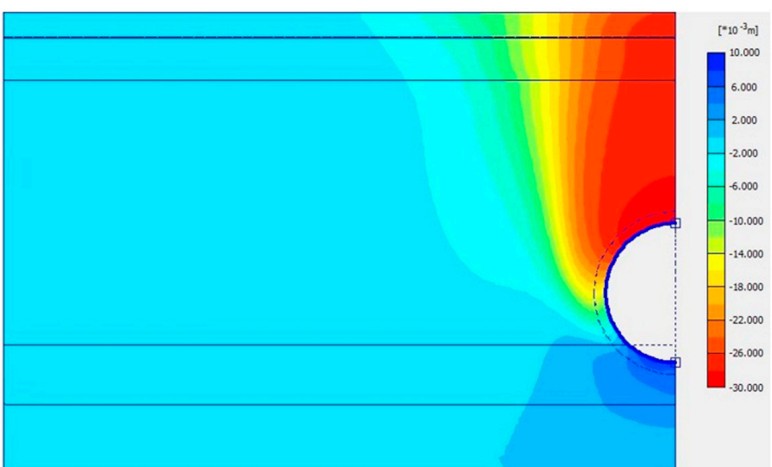

**Figure 4.** Vertical displacements in soil.

### 3.2.2. Comparison of the Modeling Methods

Figure 5 shows that the curves of the transverse settlement trough, which were yielded using the three methods, are close to the field measurements. However, the grout pressure method gives a narrower transverse settlement trough, while the contraction method offers a wider one. Finally, the proposed method provides the closest transverse settlement trough.

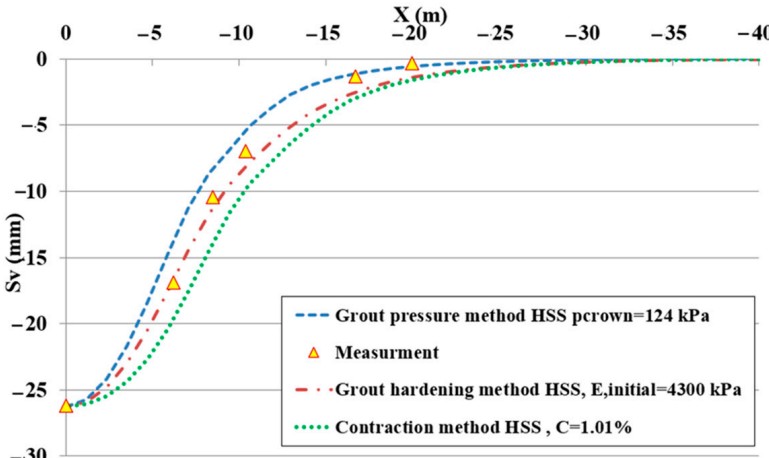

**Figure 5.** Comparison of different modeling methods.

## 4. Results and Discussion

### 4.1. Parametric Study

The purpose of the parametric study is to understand the behavior of the surface structure during the subsurface tunnel excavation and the interaction issue of tunneling, soil, and surface structure, as well as to obtain an equation to calculate the maximum settlement of the structure.

The parametric study has not included a study of the internal forces and the deformation of the tunnel lining because the contraction coefficient ($C\%$) directly represents the volume loss in the contraction method. However, the proposed method represents the volume loss indirectly through the deformation during grout pressure application and the deformation of grout with initial stiffness ($E$, initial), which are not often used in research studies. Because of this, the proposed method makes it harder to obtain and calibrate this equation. Consequently, the contraction method and the *HSS* constitutive model, which provided a good agreement with the field measurements, were adopted in the parametric study that investigated the variables of the factors shown in Figure 6, where $D$ is the tunnel diameter, e is eccentricity between the axis of the tunnel and that of the structure, $B$ is the structure width, $D_f$ is the foundation depth, $P$ is the structure load, and $H$ is the overburden depth.

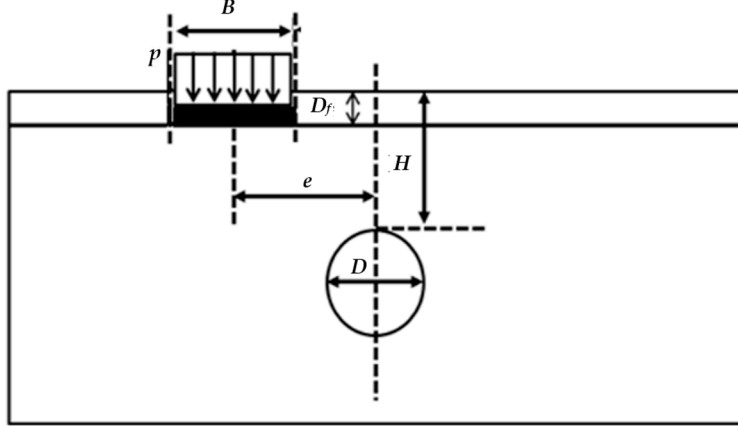

**Figure 6.** Cross section to clarify the studied parameters.

In this parametric study and for the sake of safety, the groundwater table under the greenfield condition is considered to be extremely low (27.5 m). The maximum settlement increases by 30% relative to the real case of the groundwater level depth at 1.5 m.

### 4.1.1. A Surface Structure Modeling Method

The equivalent beam method was applied to model the structure, which includes the modulus of elasticity ($E$), the moment of inertia ($I$), and the cross-sectional area ($A$) (Figure 7). This method was used by several researchers [6,26,27]. Maleki et al. [6] compared the internal forces, specifically, the bending moment of the tunnel lining, the ground surface settlement, and the ground horizontal movement profile, which had been obtained by two methods of structure modeling (e.g., real geometry model and equivalent beam model). The authors found that the results were nearly identical, so they concluded that the equivalent beam method can be used as a simple method to model the adjacent structure for practical purposes. The characteristics of the flexible beam were calculated for a structure with a number of stories ($m$), as shown in (Figure 7). It was assumed that the structure was made up of ($m + 1$) slabs with a vertical spacing between them (3.4 m). The thickness of each slab is assumed ($t_{slab}$ = 0.15 m), with dependence ($L$) in the direction perpendicular to the slab section. The following Equations (1) and (2) govern the moment of inertia and cross-sectional area:

$$I_{slab} = \frac{t_{slab}^3 L}{12} \tag{1}$$

$$A_{slab} = t_{slab} L \tag{2}$$

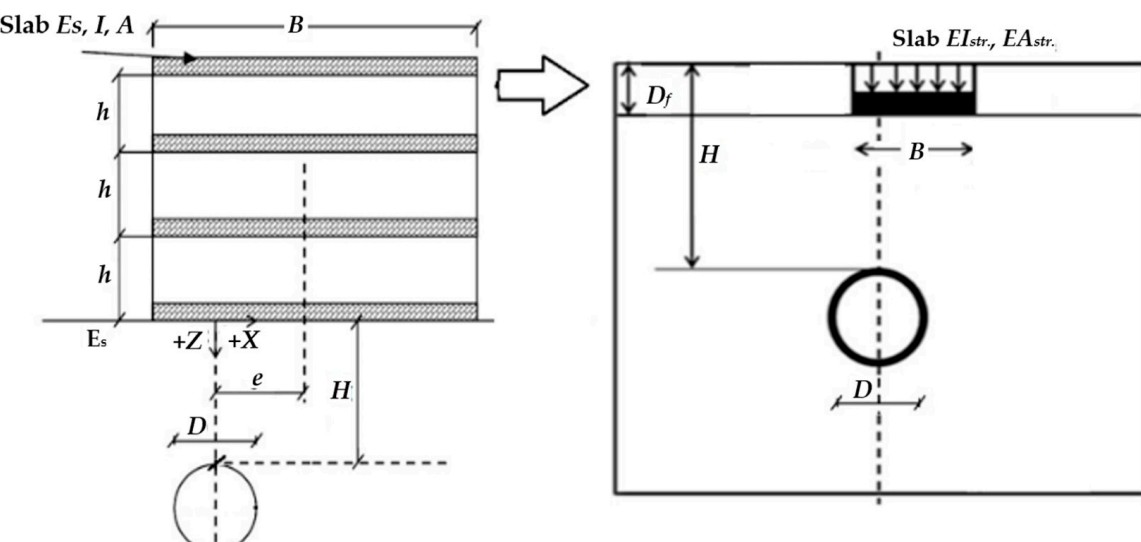

**Figure 7.** Equivalent beam method (on the basis of [6]).

In the case of plane strain, values were given ($A_{slab}$ = 0.15 m²/m, $I_{slab}$ = 0.00028 m⁴/m), as well as the elastic modulus of the concrete ($E_c$ = 23 × 10⁶ kN/m²). The equivalent beam stiffness, Equations (3) and (4), was calculated using the parallel axis theorem [28], with a neutral axis in the middle of the structure height:

$$(E_c A)_{structure} = (m + 1)(E_c A)_{slab} \tag{3}$$

$$(E_c I)_{structure} = E_c \sum_1^{m+1} \left( I_{slab} + A_{slab} h_m^2 \right) \tag{4}$$

where ($h_m$) is the vertical distance between the neutral axis of the structure and that of the slab.

The values in Table 5 were calculated, which represent the equivalent characteristics of the structures with different stories. In this study, a ten-story structure with a width of ($B$ = 13.5 m) and a uniformly distributed load of (150 kPa) was used.

**Table 5.** Equivalent structure stiffness.

| Structure | $(E_c \, I)_{structure}$ (KN·m²/m) | $(E_c \, A)_{structure}$ (KN/m) |
|---|---|---|
| 1 story | $2 \times 10^7$ | $6.9 \times 10^6$ |
| 3 stories | $2 \times 10^8$ | $1.38 \times 10^7$ |
| 5 stories | $6.96 \times 10^8$ | $2.07 \times 10^7$ |
| 10 stories | $4.39 \times 10^9$ | $3.8 \times 10^7$ |

### 4.1.2. The Influence of Structure Width ($B$)

The increase in the structure width leads to an extreme change in settlements at the end of the structure far away from the tunnel axis, where the impact of the tunneling is smaller, so an increase in structure width can reduce settlements at this end. This reduction can, in turn, impact the maximum settlement at the opposite end, closest to the tunnel axis, causing it to decrease as well. Thus, when compared to the greenfield conditions, an increase in structure width can result in a widening of the area affected by tunnel construction, as demonstrated in Figure 8. Although the maximum settlement above the tunnel axis may only experience a minor change due to an increase in structure width, there can be a significant increase in differential settlement ($\delta$) up to a ratio of structure width to tunnel diameter of approximately 2.7, as indicated in Figure 9. These findings are in agreement with earlier research conducted by Maleki [6].

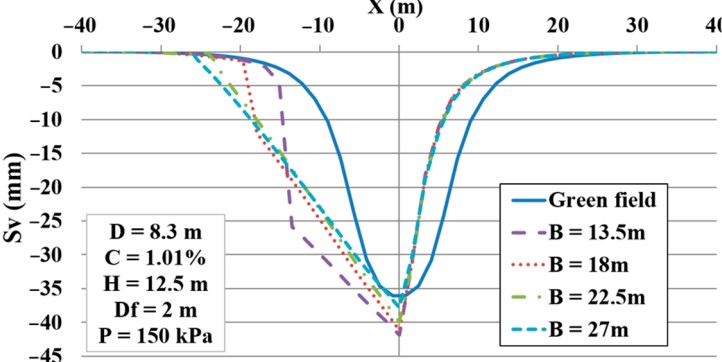

**Figure 8.** Influence of structure width on the transverse settlement trough.

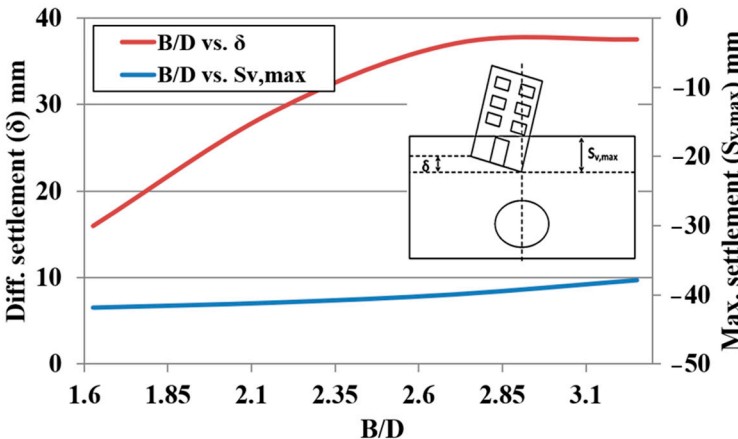

**Figure 9.** Influence of *B/D* ratio on maximum and differential settlements.

### 4.1.3. The Influence of Foundation Depth ($D_f$)

The results show that increasing the foundation depth can lead to a decrease in settlements and a change in their distribution. Specifically, maximum settlements decrease as the foundation depth increases, and the transverse settlement trough narrows as the structure approaches the top of the tunnel, as shown in Figure 10. This is due to the

decrease in soil weight above the tunnel, resulting from the increased foundation depth, which reduces the total stresses above the tunnel. However, the results also show a corresponding increase in differential settlement with increasing foundation depth, as depicted in Figure 11. The increase in differential settlement is due to the narrowing of the transverse settlement trough, which causes the building weight to affect the area directly above the tunnel and deformation to concentrate more directly above the tunnel than at the far end.

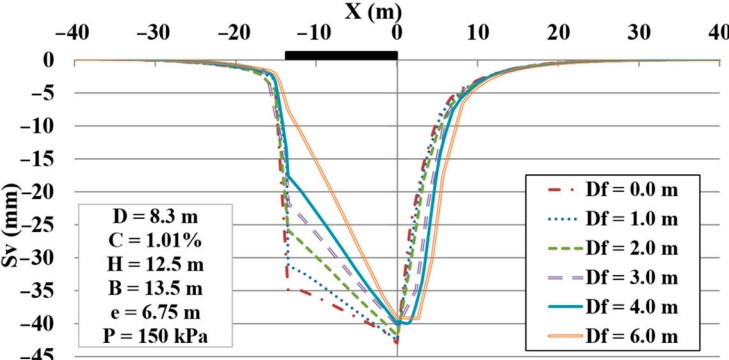

**Figure 10.** Influence of foundation depth on the transverse settlement trough.

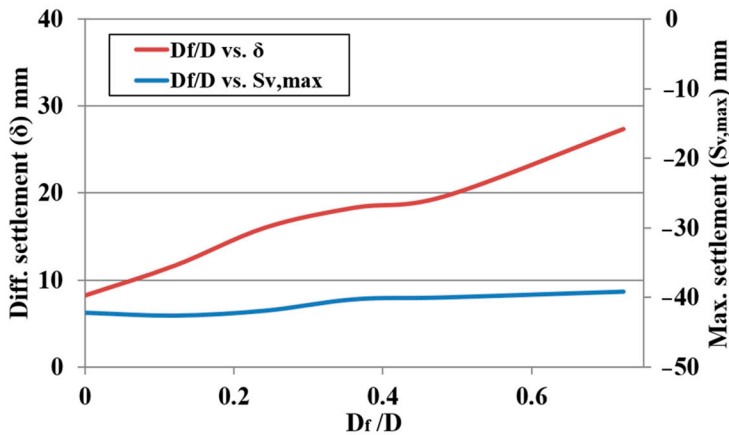

**Figure 11.** Influence of *Df/D* ratio on maximum and differential settlements.

4.1.4. The Influence of Eccentricity (*e*)

The study has uncovered the influence of eccentricity, defined as the distance between the tunnel axis and that of the structure, on the transverse settlement behavior. The study found that the size of the transverse settlement trough decreases as the eccentricity increases, while the maximum settlement above the top of the tunnel becomes less than what is observed in the greenfield conditions, as shown in Figure 12. This reduction in settlement can be attributed to the displacement caused by the presence of the structure away from the tunnel axis, which is distributed over a larger area, leading to a decrease in settlements on the top of the tunnel. Furthermore, the study has also shed light on the effect of eccentricity on the differential settlement behavior. The results demonstrate that when the eccentricity is zero, the problem is symmetrical, and the differential settlement at the structure is almost non-existent. However, as the eccentricity increases, the problem becomes asymmetric, and the maximum settlement and differential settlement increase, consistent with the results of Maleki [6]. The peak in maximum settlement and differential settlement occurs at *e/D* = 0.5, after which each of them returns to a decrease, as illustrated in Figure 13.

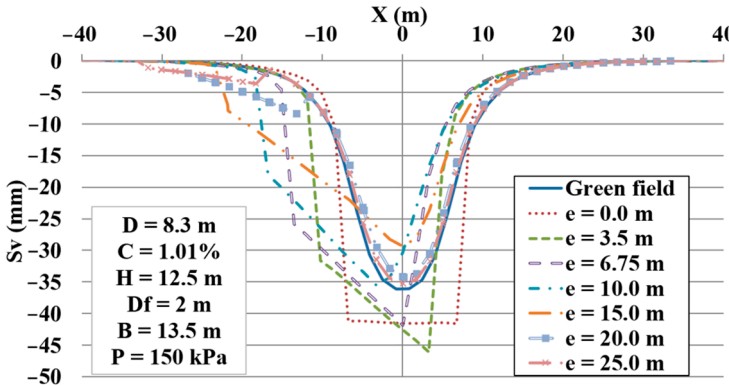

**Figure 12.** Influence of eccentricity on transverse settlement trough.

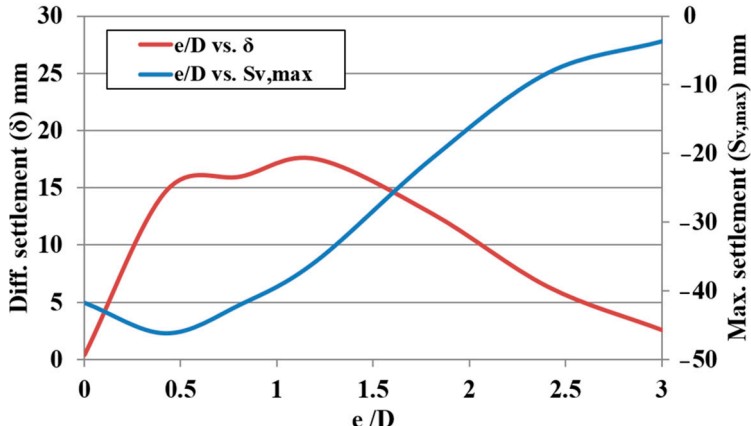

**Figure 13.** Influence of *e/D* ratio on maximum and differential settlements.

4.1.5. The Influence of Structure Load (*P*)

The study revealed that the vertical displacements between the two ends of the bottom of the structure vary less with high structure loads, resulting in a reduced differential settlement and a wider settlement trough. Conversely, when the structure load is zero, the maximum settlements change marginally in comparison with greenfield conditions, and the stiffness of the structure without any load affects only the shape of the transverse settlement trough, as depicted in Figure 14. These observations are consistent with the previous studies of Maleki [6], Law [25], and Mirhabibi [29]. Furthermore, the study investigated the impact of structure load on maximum and differential settlements. The outcomes demonstrate that increasing the structure's load leads to a greater maximum settlement and a more uniform distribution of settlement between the two ends of the structure, thereby reducing the differential settlement, as shown in Figure 15.

4.1.6. The Influence of Overburden Depth (*H*)

With the increase in the overburden depth, the results show that the maximum settlement at the bottom of the structure decreases, while the width of the transverse settlement trough increases. This behavior can be attributed to the fact that the collapse surface intersects the ground surface at a greater distance from the tunnel axis as the overburden depth increases. As a result, the values of the displacements reaching the surface decrease, and they are distributed over a larger width, as depicted in Figure 16. Moreover, the research investigated the effect of tunnel overburden depth on the maximum and differential settlements of the structure. The outcomes demonstrate that as the depth of the tunnel increases, the settlement of the structure decreases, and the differential settlement in the structure decreases, as illustrated in Figure 17.

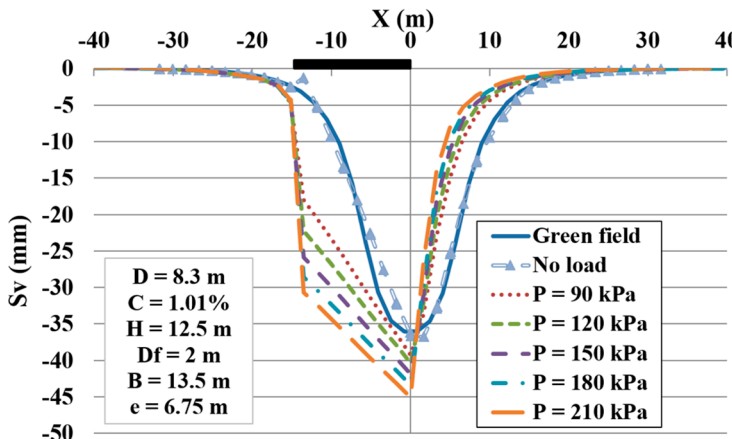

**Figure 14.** Influence of structure load on transverse settlement trough.

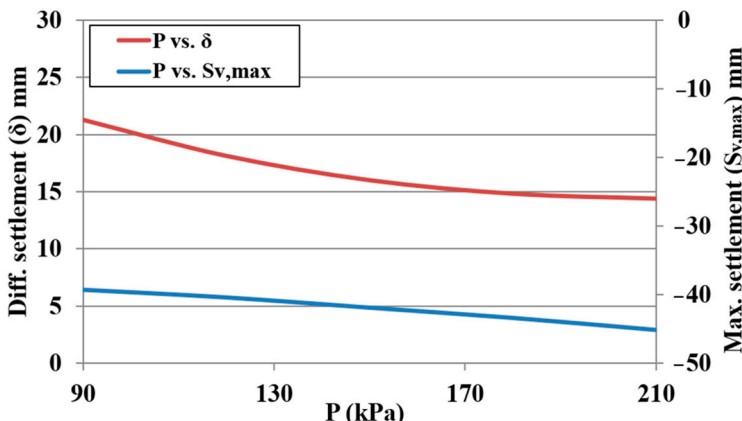

**Figure 15.** Influence of structure load on maximum and differential settlements.

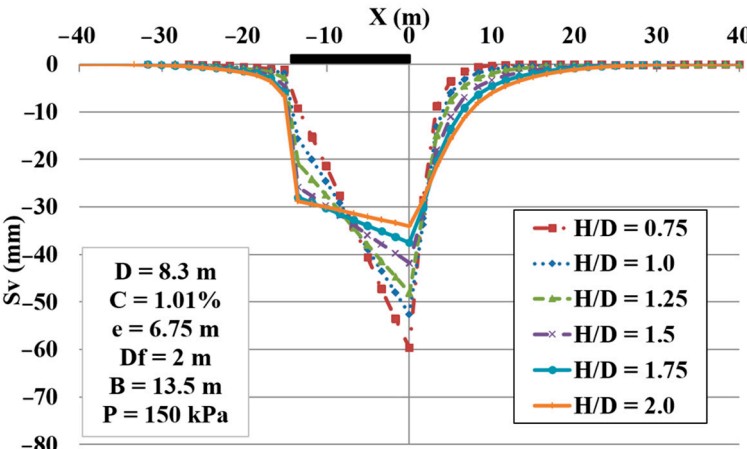

**Figure 16.** Influence of overburden depth on transverse settlement trough.

### 4.1.7. The Influence of Tunnel Diameter (*D*)

It is evident that as the diameter of the tunnel increases, the area affected by the tunnel construction expands due to the increase in deformations around the tunnel. This is mainly caused by an increase in the volume loss resulting from tunnel construction, as depicted in Figure 18. The settlement at the bottom of the structure and the dimensions of the transverse settlement trough increase as the diameter of the tunnel increases. It is also noted that the maximum settlement increases significantly, whereas the differential settlement in the structure increases slightly, due to the spread of deformations caused by the tunnel excavation over a larger area, as demonstrated in Figure 19.

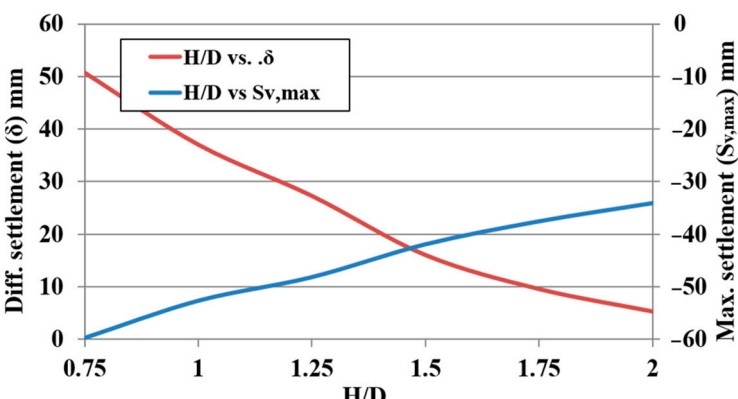

**Figure 17.** Influence of *H/D* ratio on maximum and differential settlements.

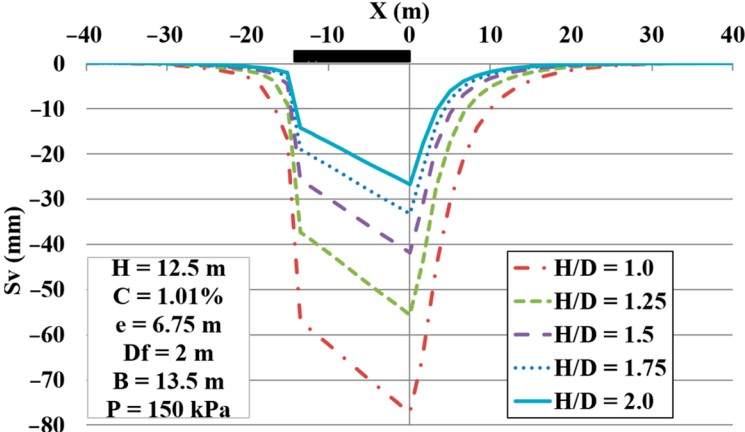

**Figure 18.** Influence of tunnel diameter on transverse settlement trough.

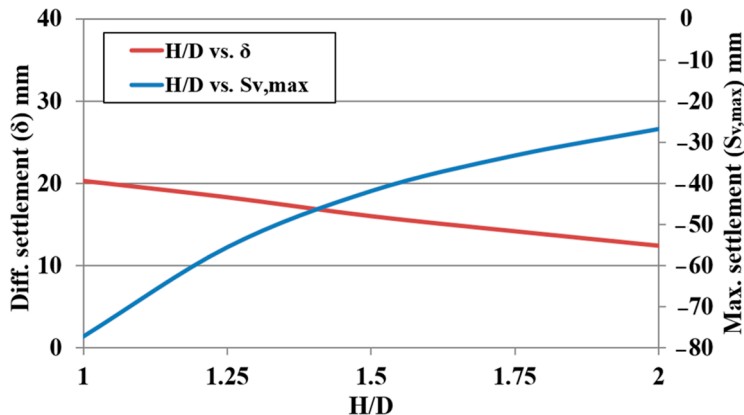

**Figure 19.** Influence of *H/D* ratio on maximum and differential settlements.

### 4.2. Equation of Elementary Analysis

A sensitivity analysis was conducted to demonstrate the influence of the various parameters on the settlements. Table 6 shows this influence through an increase of 20 % of each parameter. It is clear that the most effective parameters on the maximum settlement of the adjacent structure are the following: the tunnel diameter (*D*), the foundation eccentricity (*e*), and the overburden depth (*H*). $D_c$, means that the diameter of the tunnel is constant. Similarly, $H_c$ means that the overburden depth is constant, and *E* represents the elastic modulus of the equivalent soil layer surrounding the tunnel.

**Table 6.** The sensitivity analysis of the various parameters on the settlements.

| Parameter | $B/D$ | $D_f/D$ | $e/D$ | $P$ (kN) | $H/D_c$ | $H_c/D$ | $E$ (kPa) |
|---|---|---|---|---|---|---|---|
| **Percentage change (%)** | 20 | 20 | 20 | 20 | 20 | 20 | 20 |
| $S_{v,max}$% | 1.1 | 0.2 | 17 | 1.7 | 7.1 | 22.5 | 5 |

In this part, a newly proposed equation was developed based on the validated model of the tunnel. An extended parametric study of the independent variables ($D$, $e$, $H$, $C$), where $C$% denotes the contraction coefficient, was conducted using a total of 392 observations to derive the equation for maximum settlement (i.e., dependent variable). The presumptive values of the independent variables are listed in Table 7. The contraction method was adopted to derive the equation for the reasons explained above in Section 4.1. The EUREQA Version 1.2 software [30] was applied to find the mathematical connections between the dependent and the independent variables, based on the available database. This equation is recommended for usein the elementary analysis of tunnels.

**Table 7.** The presumptive values of the independent variables.

| $H/D$ | 0.5 | 0.75 | 1 | 1.25 | 1.5 | 1.75 | 2 | - |
|---|---|---|---|---|---|---|---|---|
| $e/D$ | 0 | 0.28 | 0.54 | 0.8 | 1.2 | 1.6 | 2 | - |
| $C$% | 0.2 | 0.4 | 0.6 | 0.8 | 1 | 1.2 | 1.4 | 1.6 |

The following Equation (5) is used to calculate the maximum settlement ($S_{v,max}$), in terms of the independent variables:

$$S_{v,max} \text{ (mm)} = 3.042 + \frac{Z}{Y + X^2 - X} - 5.91 \times C - 11.23 \times cos[(Y) + \cos(X) \times \sin(1.131 \times X) - 1.131 \times X] \qquad (5)$$

where: $Z = 17.14 \times X \times C - 59.4 \times C$, $X = \frac{e}{D}$, $Y = \frac{H}{D}$

For this analysis, the foundation width ($B$)= 13.5 m, the foundation depth ($D_f$) = 2 m, and the applied load ($P$) = 150 kPa.

Many fundamental issues should be considered with the use of this equation. This equation can be considered eligible for subsurface tunnels with the following constraints: $H/D \leq 2$, the values of the contraction coefficient should range between (0.2–1.6) %, the shallow foundation of the structure should be rigid, and the angles should be in radians.

Validation of the Proposed Equation

This paper utilized three cases to validate the proposed equation: (i) the Second Heinenoord Tunnel: a comparison between the numerical analysis outputs of the validated model and the equation solutions; (ii) the Milan Twin Tunnel: a validated model is adopted for the comparison with equation solutions; and (iii) three case studies (the Thessaloniki subway, the Naples tunnel, and centrifuge tests) are used to compare the real measurements and the equation solutions.

- The Validated Model of the Second Heinenoord Tunnel [18]

A parametric study was conducted in terms of *e/D* and *H/D* to validate the proposed equation. Figure 20 shows a consistent comparison between the numerical model results and the equation solutions.

- The Validated Model of the Milan Twin Tunnel

With an overall distance of 12.6 km, the metro-line 5 in Italy is composed of a twin tunnel with an outer diameter ($D$) = 6.7 m, and it stretches between the north and the west of Milan [31]. The twin tunnel height ($H$) = 11.65 m and the span between tunnel axes is 15 m. The tunnels were excavated in a gravely sand soil, whose properties are illustrated in Table 8. The groundwater level is 15 m, which means that the tunnels were constructed

partially underneath the water table. An EPB shield was used on site to reduce the ground movements in these highly inhabited regions. The tunnel lining was modeled as a beam element with linear elastic behavior, and the properties of the lining are listed in Table 9.

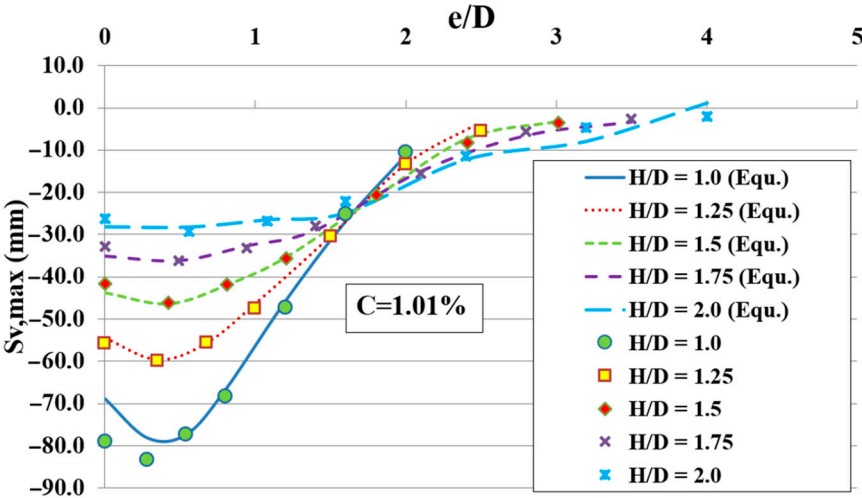

**Figure 20.** Comparison between the numerical model results and the equation solutions.

**Table 8.** The soil properties surrounding the Milan Twin Tunnel.

| $\gamma_{sat}$ (kN/m³) | $\varnothing°$ | $\psi°$ | $c$ (kPa) | $E_{50}^{ref}$ (kN/m²) | $E_{oed}^{ref}$ (kN/m²) | $E_{ur}^{ref}$ (kN/m²) | $m$ | $G_0^{ref}$ (kN/m²) | $\gamma_{0.7}$ (%) | $v_{ur}$ | $R_{inter}$ |
|---|---|---|---|---|---|---|---|---|---|---|---|
| 20 | 33 | 0 | 0 | 48,000 | 48,000 | 144,000 | 0.4 | 250,000 | 0.0001 | 0.2 | 0.67 |

**Table 9.** The properties of the Milan Twin Tunnel lining.

| Parameter | $t$ (m) | $\gamma$ (kN/m³) | $v$ | $E$ (GPa) |
|---|---|---|---|---|
| **Tunnel lining** | 0.3 | 25 | 0.15 | 35 |

E: Young's modulus.

The field measurements of settlements of the transversal ground section S16 under the greenfield conditions were conducted after the excavation of the first tunnel and both tunnels, respectively. Plaxis 2D software was adopted in this study to validate the twin tunnel (for additional information, see [31]).

*HSS* was employed to simulate the behavior of the surrounding soil of the twin tunnel. The contraction coefficients are depicted in Figure 21. This study [31] targeted the ground surface settlements that resulted from the excavation of the first tunnel.

Based on the back analysis method, a comparison was performed between the measured surface subsidence in the field and those predicted by the *HSS* model. The field measurements were performed in two stages: (1) after the complete excavation of the first tunnel, and (2) after the excavation of the second tunnel. Figure 21 indicates a good agreement between the field data and the *HSS* results.

After the validation of this model, a structure represented by a distributed load (*P*) of 150 kPa was added to the model, where the foundation width (*B*) was 13.5 m and the foundation depth (*D*f) was 2 m. Figure 22 illustrates the comparison between the proposed equation results and the numerical model outputs for the maximum settlements of the structure. The presented equation results are relatively compatible with those of FEM.

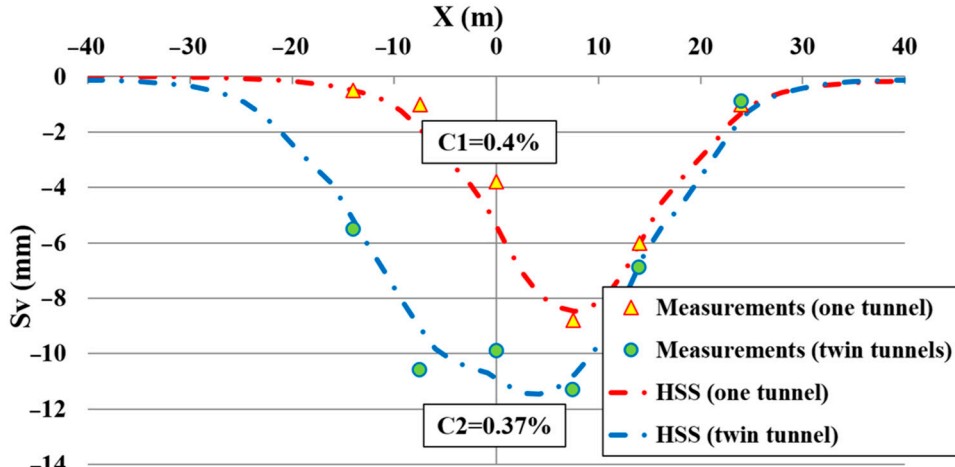

**Figure 21.** Comparison between the field measurements and the results of numerical modeling using *HSS* constitutive model of the Milan Twin Tunnel.

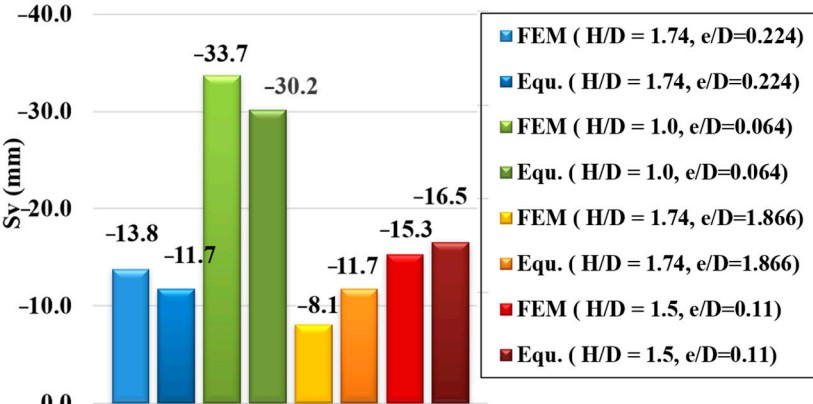

**Figure 22.** Comparison between the proposed equation results and the numerical model outputs for the maximum settlements of the structure.

- Case Studies
  - The Thessaloniki subway, Greece

The northbound and southbound lines are the designations of the twin tunnels that make up the Thessaloniki city subway in Greece [32]. The two lines pass through densely populated areas at relatively shallow depths, ranging from 8 to 20 m, and are in proximity to adjacent structures. Two TBM-EPBs were utilized to decrease the surface settlements when the excavation passed through a sandy clay deposit. The groundwater table is located 5 meters beneath the ground surface. Settlements of the stiff raft foundation of the adjacent structure D91 were measured after the excavation of the southbound tunnel (first executed tunnel), which consists of seven stories. The site conditions and the design parameters are as follows: the tunnel diameter ($D$) = 6.2 m, the overburden depth ($H$) = 14.5m, the eccentricity ($e$) = 22.5 m, the raft width ($B$) = 11 m, the over excavation (20 mm), and the contraction coefficient ($C$) = 0.65%.

Figure 23 shows a comparison between the proposed equation and the field measurements for the maximum settlements of the structure raft foundation. The result of the present equation is somewhat consistent with the measured settlement.

  - Line 6 of the Naples Underground

The Naples Underground Line 6 is a part of the public railway network provided by the Municipal Plan of Transportation for the Metropolitan Area of Naples (Italy) [33]. The Santa Maria Della Vittoria church with a width of 17.3 m is located next to the path

of the Line 6 tunnel, which was excavated between 2009 and 2011. Field measurements were conducted on the front façade of the church. The overburden depth ($H$) = 13.225 m, the tunnel diameter ($D$) = 8.15 m, and the eccentricity ($e$) = 12.6 m, while the contraction coefficient was ($C$) = 0.3%. The equation gives a result that is fairly close to the field measurements of the maximum settlement of the church, see Figure 23.

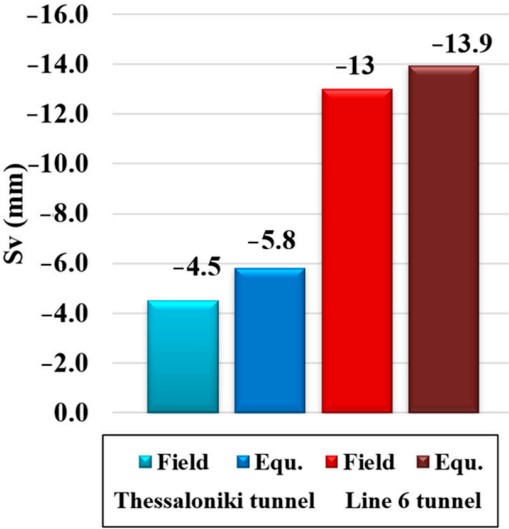

**Figure 23.** Comparison of the measured and calculated values of the two case studies.

- Comparison Based on the Centrifuge Tests

The results of the centrifuge test series for the tunnel in regular dense dry silica sand performed by Ritter et al. [34,35], at 75 g, were considered. A structure on strip footings affected by tunneling was tested in 1/75th scale models. The dimensions corresponding to the prototype scale (1/75th) are as follows: the tunnel diameter ($D$) = 6.2 m, the cover-to-diameter ratio ($H/D$) = 1.3, and the overburden depth ($H$) = 8.2 m. The contraction coefficient is ($C$) = 1% [36]. Table 10 shows the parameters used for each test. Figure 24 confirms that the equation results are in good agreement with the results of the centrifuge test.

**Table 10.** The parameters used for each test (compiled by the authors after Franza et al., 2020 [36]).

| Test Number | $B$ (m) | $e$ (m) | $H/D$ | $e/D$ |
|---|---|---|---|---|
| 1 | 15 | 12 | 1.3 | 1.935 |
| 2 | 19.5 | 9.75 | 1.3 | 1.573 |

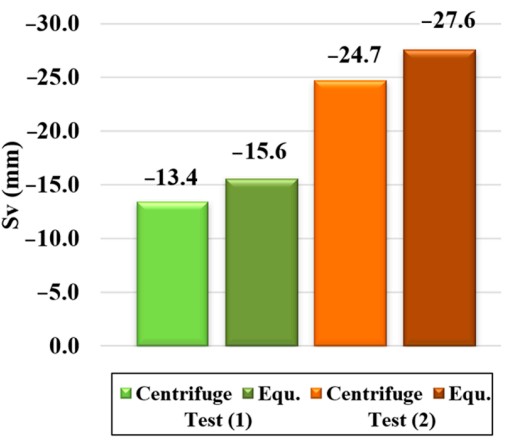

**Figure 24.** Comparison of the measured and the calculated values of the centrifuge tests.

### 5. Conclusions

Tunneling is a critical aspect of infrastructure development, with accuracy and efficiency being essential for ensuring the safety and stability of surrounding structures. This study presents a more realistic 2D modeling method that predicts ground movement and settlements accurately. Additionally, an extensive parametric study was conducted to examine how the presence of adjacent structures impacts the deformation resulting from tunneling, in comparison with deformation under greenfield conditions. A new equation for calculating maximum settlement under adjacent structures simplifies the design process and provides more accurate predictions, leading to safer and more efficient infrastructure development.

Plaxis 2D software was used to investigate the interactions of a tunnel–soil–structure system. The following conclusions can be drawn:

1. A new and more realistic modeling method is proposed for simulating the tunnel excavation process. Although the reference modeling methods, namely the contraction method and grout pressure method, have provided settlement troughs that were fairly close to the measured value, the grout hardening method proposed in this study has demonstrated a greater level of consistency with the field measurements.
2. The Hardening Soil Model with Small-Strain Stiffness (*HSS*) has yielded a transverse settlement trough closer to the field measurement than that of the Hardening Soil model (*HS*).
3. Increasing structure width, foundation depth, and overburden depth reduces the maximum settlement under the structures. In contrast, the structure load and tunnel diameter have negatively affected the maximum settlements.
4. Given the different values of the eccentricity between the tunnel and the structure, the maximum settlements increase as the ratio (e/D) increases to be 0.5, and then decrease.
5. Each parameter studied has a unique impact on the settlement behavior. In addition, the tunnel diameter, overburden depth, and eccentricity have a noticeably greater effect on the settlement at the ground surface.
6. The increase in the structure width and the foundation depth has corresponded to an increment in the structure differential settlements, whilst the structural load, the overburden depth, and the diameter have reduced the differential settlements to varying degrees.
7. Based on the finite element method (FEM), a new equation is developed for the elementary analysis of the tunnels that calculates the maximum settlements beneath the adjacent structures. A comparison of the results of the proposed equation with the field measurements has been conducted for several case studies. The results of the current equation are in good agreement with the measured values.

The proposed method and equation can serve as valuable tools for engineers and researchers in the field of tunneling and infrastructure development, enabling them to make informed decisions and minimize the risks associated with these complex projects. The contributions of this study are expected to have a significant impact on the field of tunneling and to pave the way for future research and development in this critical area of infrastructure engineering.

**Author Contributions:** Conceptualization, A.S., R.A. and A.A.; proposed design method, A.S. investigation, A.A.; writing—original draft preparation, R.A. and A.A.; numerical modeling, R.A., A.S. and A.A.; writing—review and editing, A.S., R.A. and A.A. All authors have read and agreed to the published version of the manuscript.

**Funding:** This research received no external funding.

**Data Availability Statement:** Data will be made available upon request.

**Conflicts of Interest:** The authors declare no conflict of interest.

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
