# Peer review of "Two-Dimensional Numerical Analysis for TBM Tunneling-Induced Structure Settlement: A Proposed Modeling Method and Parametric Study"

_infrastructures, doi:10.3390/infrastructures8050088_

Round 1

Reviewer 1 Report

An interesting paper which presents the two dimensional finite element analyses and parametric studies of settlement during tunneling process. Please consider the following comments while preparing the revised manuscript:

1. Though the work is unique, the discussions pertaining to the important findings are very weak and requires significant improvement in the revised version of the paper.

2. Abstract: The length of the abstract is to be reduced within the word limit of 200. Remove redundant data and present only important findings/observations. Some quantitative results are also to be added to the abstract.

3. Why the authors proposed a 2D numerical modeling rather than a 3D modeling which is much accurate to represent the real-time behavior?

4. Section 3.1: What is the proof/reference for the data presented in the present section.

5. Table 1: Does the values are subjected to change at different conditions? Any standard deviations required? Please clarify.

6. Did the authors perform any mesh sensitivity analysis?

7. Please clarify the accuracy of the obtained predictions from the 2D numerical analysis.

8. Please clarify the rationale behind the selection of parameters in the parametric studies.

9. Conclusion part of the paper is to be significantly improved to reflect the actual findings of the proposed work.

Paper can be accepted following the minor revision

Author Response

Reviewer /1/:

Thank you for your cooperation.

Reviewer point #1: Though the work is unique, the discussions pertaining to the important findings are very weak and requires significant improvement in the revised version of the paper.

Author response #1:  We carefully reviewed our discussions and findings and make the necessary revisions to improve the clarity and strength of our arguments. We also added more figures to further clarify the behavior of the system under different conditions.

Reviewer point #2: Abstract: The length of the abstract is to be reduced within the word limit of 200. Remove redundant data and present only important findings/observations. Some quantitative results are also to be added to the abstract. 

Author response #2: The authors have abbreviated and rewritten the abstract.

Reviewer point #3: Why the authors proposed a 2D numerical modeling rather than a 3D modeling which is much accurate to represent the real-time behavior?

Author response #3: Several reference studies, confirmed the convergence between the results of the two- and three-dimensional numerical analysis. Where the issue of tunneling is considered a plane strain problem and the adoption of a three-dimensional modeling of a plane strain problem (homogeneous soil, the absence of geographic locations or changes in the longitudinal direction parallel to the tunnel axis) requires additional effort and time without adding to the accuracy of the results, and also the aim of our research is not to study the evolution of a trough Subsidence with the progress of tunneling, but rather the effect of the final subsidence trough on the structure above the tunnel and the effect of the presence of the structure on the subsidence trough compared to the case of the green field.

Therefore, we opted for 2D modeling as a more practical and efficient approach while still ensuring the accuracy and precision of the results through validation and calibration with field measurements. We believe that this approach allowed us to conduct a comprehensive parametric study while minimizing computational resources and time.

Reviewer point #4: Section 3.1: What is the proof/reference for the data presented in the present section

Author response #4:  We added it.

[18] Van Jaarsveld, E.P.; Plekkenpol, J.W. Ground Deformations due to the Boring of the Second Heinenoord. Geotechnical Engi- 495 neering for Transportation Infrastructure - Theory and Practice: Proceedings of the Twelfth European Conference on Soil Me- 496 chanics and Geotechnical Engineering, Amsterdam, 1999, 153-159. DOI:10.1201/9781439834268.CH3  

Reviewer point #5: Table 1: Does the values are subjected to change at different conditions? Any standard deviations required? Please clarify.

Author response #5:  We wish to clarify that the data presented in Table 1 are field-measured values that were determined under specific conditions and are employed to validate our model. As provided in the original source, we have presented the data in Table 1 without any accompanying standard deviation values.

We would like to emphasize that since the values presented in Table 1 correspond to actual measurements taken in the field and are not model-generated estimates or predictions, it is unnecessary to calculate standard deviations to account for any potential variability in the results.

Reviewer point #6: Did the authors perform any mesh sensitivity analysis?

Author response #6: We wish to confirm that we have performed a mesh sensitivity analysis. Our findings suggest that a coarse mesh with refinement around the tunnel and building provides accurate results that are consistent with the field measurements.
We believe that mesh sensitivity analysis is an important aspect of the numerical modeling process, as it ensures that the results of the model are not sensitive to the mesh size and provides confidence in the accuracy of the results. We have provided a description of our mesh sensitivity analysis in the manuscript to document our methodology and to assist readers in reproducing our results.

Author response #7: Please clarify the accuracy of the obtained predictions from the 2D numerical analysis.

Reviewer point #7:  Our study has shown that the 2D numerical model provides accurate predictions of the ground settlements induced by the tunnel construction.

To assess the accuracy of our model, we have compared the predicted results with field measurements collected during the construction of the tunnel. Our findings demonstrate that the predictions from the 2D numerical model are in good agreement with the field measurements, with a reasonable level of accuracy.

Overall, our study shows that the 2D numerical model can provide accurate predictions of the ground settlements induced by the tunnel construction, and it can be a valuable tool for assessing the potential impact of tunnel construction on adjacent structures.

Reviewer point #8: Please clarify the rationale behind the selection of parameters in the parametric studies.

Author response #8: Our selection of parameters was based on the objectives and scope of our study, which is to investigate the influence of key parameters on the ground settlement induced by the tunnel construction.

We have identified several key parameters that are known to affect the ground settlement, based on the literature review and expert knowledge. These parameters include the tunnel depth, tunnel diameter, distance between the tunnel and the building, stiffness of the building foundation, soil properties, and building weight.

To investigate the influence of these parameters on the ground settlement, we have designed a set of parametric studies where we vary one parameter at a time while keeping other parameters constant. We have selected a range of values for each parameter that are relevant to the typical conditions encountered in practice.

Overall, our selection of parameters in the parametric studies was based on a careful consideration of their importance in influencing the ground settlement induced by the tunnel construction and their relevance to the research questions or objectives. We believe that the results of our parametric studies provide valuable insights into the influence of these key parameters on the ground settlement and can be used to optimize the design and construction of tunnels in urban areas.

.

Reviewer point #9: Conclusion part of the paper is to be significantly improved to reflect the actual findings of the proposed work.

Author response #9: We acknowledge that the conclusion section needs improvement to better reflect the findings of our work. In the revised version of the paper, we revised the conclusion section and provided a more detailed summary of the results and their significance. We also ensured that our conclusions are fully supported by the findings presented in the paper.

Reviewer 2 Report

2D Numerical Analysis for TBM Tunnelling-Induced Structure

Settlement:    A Proposed Modelling Method and Parametric Study

Rashad Alsirawan, Ashraf Sheble and Ammar Alnmr

This paper introduces a practical TBM excavation simulation method. A validated TBM-excavated Second Heinenoord tunnel model with the slurry shield compares the suggested method to other reference modelling methods. In addition, tunnel excavation causes structural settlements. Finally, the study examines how structure breadth, foundation depth, eccentricity, structure load, overburden depth, and tunnel diameter affect tunnel-soil-structure interaction.

The study concludes with an essential parameter-based equation for estimating the ultimate level of structure settlement.

The accuracy of the equation is checked by comparing it to a recently verified model, two real-world case studies, and centrifuge tests. Good agreement between predicted and measured settlements indicates settlement behavior caused by TBM tunneling. This new simulation approach and parametric study can assist guarantee the structural integrity of TBM tunneling projects in urban areas.

Comments:

The title of the article is clear, it is consistent with the main body of the text, and it is pertinent to the topic that is being researched. The study that establishes the connection between the stated problem and earlier articles that dealt with similar themes is connected to the form in such a way that it creates a relationship between the two. Because it touches on a generally intriguing subject, the article generally deals with an interesting issue, so I recommend publishing it after minor corrections.

Line 24

Finally, the paper proposes an equation for predicting the maximum settlement of a structure, taking into consideration the most critical parameters.

rewrite for clarity

Finally, the paper proposes an equation for predicting a structure's maximum settlement, considering the most critical parameters.

Line 173

A less precise model may yield poor results and make the estimation of the true behaviour in the field more difficult [24].

rewrite to

A less precise model may yield poor results and make estimating the proper behaviour in the field more complex [24].

Line 192

However, the grout pressure method gives a narrower transverse settlement trough, while the contraction method offers a wider transverse settlement trough.

rewrite to

However, the grout pressure method gives a narrower transverse settlement trough, while the contraction method offers a wider one.

Line 203

The parametric study has not included a study of the internal forces and the deformation of the tunnel lining because the volume loss in the contraction method is directly represented by the contraction coefficient.

rewrite to

The parametric study has not included a study of the internal forces and the deformation of the tunnel lining because the contraction coefficient directly represents the volume loss in the contraction method.

Line 205

However, the proposed method represents the volume loss indirectly through the deformation during grout pressure application and the deformation of grout with initial stiffness (E, initial), which are not widely used in research studies; thus, the proposed method complicates the possibility of obtaining and calibrating this equation.

replace with

However, the proposed method represents the volume loss indirectly through the deformation during grout pressure application and the deformation of grout with initial stiffness (E, initial), which are not often used in research studies. Because of this, the proposed method makes it harder to get and calibrate this equation.

Line 214

In this parametric study and for the sake of safety, the groundwater table under the greenfield condition is considered to be extremely low (27.5 m). The maximum settlement increases by 30% relative to the real case of the groundwater level depth with 1.5 m.

replace with

In this parametric study and for safety, the groundwater table under the Greenfield condition is considered extremely low (27.5 m). The maximum settlement increases are 30% relative to the actual case of the groundwater level depth of 1.5 m.

Line 273

The study has also demonstrated that the differential settlement at the structure is almost non-existent in the case of the eccentricity being equal to zero. Nevertheless, as eccentricity increases, the maximum and the differential settlement both increase slightly, followed by a decrease, which is consistent with the results of Maleki [6].

replace with

The study has also demonstrated that the differential settlement at the structure is almost nonexistent if the eccentricity is equal to zero. Nevertheless, as eccentricity increases, the maximum and the differential settlement increase, followed by a decrease, which is consistent with the results of Maleki [6].

Line 300

remove which is

Line 335

This equation can be considered to be eligible for the subsurface tunnels, H/D ≤ 2, the values of the contraction coefficient should range between (0.2-1.6) %, the shallow foundation of the structure should be rigid, the angles should be in Radians).

replace with

For example, this equation can be considered eligible for the subsurface tunnels: H/D ≤ 2, the values of the contraction coefficient should range between (0.2-1.6) %, the shallow foundation of the structure should be rigid, and the angles should be in radians.

Line 373

Figure 15 shows a good agreement between the field measurements and the results, which were yielded by HSS. 

rewrite to

Figure 15 indicates a good agreement between the field data and the HSS results. 

Line 423

Move the description of Figure 18 below the picture on page 15

Line 425

In this study, the authors have used the Plaxis 2D connect edition ver 20 software to investigate the interactions of a tunnel-soil structure system. To simulate the excavation progress, a more realistic modelling method of four stages has been utilized.

rewrite to

The authors have used the Plaxis 2D Connect Edition ver. 20 software to investigate the interactions of a tunnel-soil structure system. A more realistic four-stage modelling method has been implemented to simulate the excavation progress.

Line 435

3- The increase of the structure width, the foundation depth, and the overburden depth has contributed to reducing the maximum settlement under the structures, while the structure load and the tunnel diameter have had a negative effect on the maximum settlements.

rewrite to

3- Increasing structure width, foundation depth, and overburden depth have reduced the maximum settlement under the structures. In contrast, the structure load and tunnel diameter have negatively affected the maximum settlements.    

Line 440

4- Given the different values of the eccentricity between the tunnel and the structure, the maximum settlements have increased as the ratio (e/D) increased to be 0.5, and then it started to decrease. 

rewrite to

4- Given the different eccentricity values between the tunnel and the structure, the maximum settlements increased as the ratio (e/D) increased to 0.5, and then decreased.

Line 448

The results of the current equation seem to be in good agreement with the measured values.

rewrite to

The results of the current equation are in good agreement with the measured values.

 Minor English language editing is required (such as definite and indefinite articles, etc.)

Author Response

The authors have made the suggested corrections based on your feedback, and I would like to express my gratitude for your input.

Reviewer 3 Report

1.      L76-77: the clarity of the sentence need to be checked

2.      N L64- 90, the format of reporting the works done by the authors should be checked to reduce the use of “has”, and “have”. This could be for example in L80 for example, “O’Reilly & New[14]… proposed an equation… “

3.      After L90, a general summary is needed to capture the inadequacies in the previous works reviewed and how the proposed study w\is different. Summarily, I suggest the authors filled the gaps between the switch in sentences in L91 and L92.

4.      In L98, under the section 2, I suggest the authors reconsider the opening sentence, with the passive tone “The authors....” and replace it with a more active sentence.

5.      In L99 – L107, the statements are having a disconnecting flow as the model phases are not properly explained in an interconnected manner.  

6.      In figure 1, I suggest the authors provide a hierarchical connection to show that the phases are in relation to one another. Also, I suggest the authors consider using arrows pointer to differentiate the parameters used in the illustrations and the whole graphical illustrations should be remade.

7.      The explanations as contained in L110-124 should be properly explained in appropriate subheadings, also the use of grammar and punctuation marks should be checked. The statements as contained within this lines are lacking clarity thus making it difficult to understand the methodology. The authors should also check, there were no explanations on the fourth phase.

8.      In Section 3 which describe the finite element modelling, there should be previous discussion on what the section is about before delving into the explanation of the section topic.

9.      The referenced work in which the contents of Table 1 was extracted from need to be cited.

10.  I suggest the authors check through for the use of “has/ has been” and have/ have been” in the whole paper. There are many cases of inappropriate usage.

11.  Additionally, I suggest the authors while referring to cases as contained in L403 for an example should make mention of the names rather than a pointer.

12.  I suggest the authors present the methodology of their approach in a more coherent and logical format, and applying them under subsections.

13.  The conclusions aspects from L424 should dwell also on the applicability of the study more than a summary of what the study is about. Also, the conclusions should be connected so as to provide a strong take home message for the readers. 

Minor editing of English language required

Author Response

Reviewer /3/:

Thank you for your cooperation.

Reviewer point #1: L76-77: the clarity of the sentence need to be checked. 

Author response #1: We agreed with the reviewer, it is modified.

Boldini [12] conducted an investigation to demonstrate the impact of the number of storeys on the settlement trough. Specifically, the study examined how the incremental stiffness of the structure with more storeys affects the settlement trough, and also investigated the effect of the incremental structure's weight on the settlement trough.

Reviewer point #2: N L64- 90, the format of reporting the works done by the authors should be checked to reduce the use of “has”, and “have”. This could be for example in L80 for example, “O’Reilly & New[14]… proposed an equation… “. 

Author response #2: We agreed with the reviewer, it is modified.

Reviewer point #3: After L90, a general summary is needed to capture the inadequacies in the previous works reviewed and how the proposed study w\is different. Summarily, I suggest the authors filled the gaps between the switch in sentences in L91 and L92. 

Author response #3: We agreed with the reviewer, The authors modified this part. We added and modified the following:

The majority of previous studies have primarily focused on the analysis of settlement through greenfield conditions. Furthermore, the parametric study has been limited in its ability to fully comprehend the behavior of settlement under structures, and the proposed equations used to calculate settlements have been primarily based on green field conditions. This highlights the need for further research to better under-stand the complexities of settlement behavior under various conditions, including those that arise when structures are present.

In this paper, a validated model of the subsurface Second Heinenoord tunnel [18] is utilized to calibrate a new and improved modelling method of shield tunnelling using the Finite Element Method (FEM). The proposed method aims to provide a more accurate representation of the tunnelling process in the real world. Additionally, an extensive parametric study is conducted to explore the impact of various critical parameters on the settlement trough under adjacent structures. Finally, a novel and tailored equation is proposed that can be applied in the elementary analysis to calculate the maximum settlements of adjacent structures.

Reviewer point #4: In L98, under the section 2, I suggest the authors reconsider the opening sentence, with the passive tone “The authors....” and replace it with a more active sentence. 

Author response #4: We modified this part.

Reviewer point #5: In L99 – L107, the statements are having a disconnecting flow as the model phases are not properly explained in an interconnected manner.

Author response #5: The authors have made every effort to revise this section. Following a thorough and precise revision process, we are hopeful that it has been able to effectively address the intended perspectives.

Reviewer point #6: In figure 1, I suggest the authors provide a hierarchical connection to show that the phases are in relation to one another. Also, I suggest the authors consider using arrows pointer to differentiate the parameters used in the illustrations and the whole graphical illustrations should be remade. 

Author response #6: Initially, we attempted to use arrows in the figure, but found that it was not very clear. We made changes to the presentation in order to improve clarity. The figure displays the interface of the Plaxis 2D, and we chose this method of representation for ease of understanding. However, we are open to suggestions for further improvement. If you have any suggestions, please do share them with us with one example for more clarity.

Reviewer point #7: The explanations as contained in L110-124 should be properly explained in appropriate subheadings, also the use of grammar and punctuation marks should be checked. The statements as contained within these lines are lacking clarity thus making it difficult to understand the methodology. The authors should also check, there were no explanations on the fourth phase.

Author response #7: The authors explain here the steps (stages) should be made in the staged construction tool in Plaxis. After the correct construction of common tunnels. These steps guide the designer what should he/she do to apply this method of modeling and get more reliable representation of the tunnel excavation. Based on that, no more details provide them. We appreciate your feedback and suggestions, and we are open to receiving more feedback to help develop and improve our work. We make the mentioned corrections, thank you very much.

Reviewer point #8: In Section 3 which describe the finite element modelling, there should be previous discussion on what the section is about before delving into the explanation of the section topic.

Author response #8: We added it for more clarification for readers

In order to validate the proposed modeling method, a case study of the sub-surface Second Heinenoord tunnel [18] was utilized. This section of the study involves two key aspects: (i) conducting a 2D numerical modeling of the tunnel using the conventional contraction method to determine the appropriate constitutive model, by comparing the predictions of numerical analysis using the Hardening Soil with Small-strain Stiffness (HSS) and the Hardening Soil (HS), and (ii) comparing the results obtained from three distinct modeling methods (grout pressure method, grout hardening method, and contraction method) with field measurements, with the aim of validating the proposed method.

Reviewer point #9: The referenced work in which the contents of Table 1 was extracted from need to be cited.

Author response #9: We added it

Reviewer point #10: I suggest the authors check through for the use of “has/ has been” and have/ have been” in the whole paper. There are many cases of inappropriate usage.

Author response #10: Thank you very much for this notification. We modified them and we will ask the editor for new proofreading.

Reviewer point #11: Additionally, I suggest the authors while referring to cases as contained in L403 for an example should make mention of the names rather than a pointer.

Author response #11: We modified the figures; we kindly ask the reviewer for more explanation if the modification is not satisfying

Reviewer point #12: I suggest the authors present the methodology of their approach in a more coherent and logical format, and applying them under subsections.

Author response #12: We appreciate your suggestion to present our methodology in a more coherent and logical format with subsections. We have presented our methodology throughout our paper (generally in the introduction of each section), detailing the steps we took in each section to address the research questions and objectives. While we understand the benefits of having a separate methodology section, we decided to present our methodology within each section to avoid repeating the same information multiple times. We have organized the sections carefully to ensure that our methodology is clearly and thoroughly explained

Reviewer point #13: The conclusions aspects from L424 should dwell also on the applicability of the study more than a summary of what the study is about. Also, the conclusions should be connected so as to provide a strong take home message for the readers.

Author response #13: We modified this part and tried to make balance between all the comments of the reviewers

Reviewer 4 Report

This article presents 2D numerical analysis for TBM tunnelling-induced structure settlement. The manuscript is very well organized and the methods are correct.

I recommend acceptance of this manuscript subject to a minor revision.

1. The authors should specify the geographic locations (such as hills, desert, planes etc.) where the application of this structure will be most appropriate, accurate and effective.

2. What are limitations of this analysis?

3. What are the existing research gaps and how does this current research fill them?

The quality of English language is fine.

Author Response

Reviewer /4/:

Thank you for your cooperation.

Reviewer point #1: The authors should specify the geographic locations (such as hills, desert, planes etc.) where the application of this structure will be most appropriate, accurate and effective. 

Author response #1: A problem that achieves the plane-strain conditions was selected in order to use a two-dimensional analysis; the surface of the ground is horizontal in our problem as shown in Figure 2, the authors added this sentence to clarify this point “the ground surface is horizontal”. In the case of existence geographic locations, a three-dimensional analysis is needed to represent the problem

Reviewer point #2: What are limitations of this analysis? 

Author response #2: We would like to clarify that the proposed modeling method has no inherent limitations, as its effectiveness depends on the calibration and validation processes. To provide additional context, we have included information to highlight that the case study pertains to a subsurface tunnel. In addition, the limitations of the proposed equation in the last part are exist.

Reviewer point #3: What are the existing research gaps and how does this current research fill them? 

Author response #3: We added this part to the introduction

The majority of previous studies have primarily focused on the analysis of settlement through greenfield conditions. Furthermore, the parametric study has been limited in its ability to fully comprehend the behavior of settlement under structures, and the proposed equations used to calculate settlements have been primarily based on green field conditions. This highlights the need for further research to better under-stand the complexities of settlement behavior under various conditions, including those that arise when structures are present.

In this paper, a validated model of the subsurface Second Heinenoord tunnel [18] is utilized to calibrate a new and improved modelling method of shield tunnelling using the Finite Element Method (FEM). The proposed method aims to provide a more accurate representation of the tunnelling process in the real world. Additionally, an extensive parametric study is conducted to explore the impact of various critical parameters on the settlement trough under adjacent structures. Finally, a novel and tailored equation is proposed that can be applied in the elementary analysis to calculate the maximum settlements of adjacent structures.

Round 2

Reviewer 3 Report

The authors have satisfactorily addressed the comments raised by the reviewers.